# Using a theory-informed approach to explore patient and staff perspectives on factors that influence clinical trial recruitment for patients with cirrhosis and small oesophageal varices

Clair Le Boutillier[1]*, Claire Snowdon[2], Vishal Patel[3,4,5], Mark McPhail[3,4], Christopher Ward[6], Ben Carter[1], Ruhama Uddin[3], Ane Zamalloa[3], Vanessa Lawrence[1]

1 Institute of Psychiatry, Psychology & Neuroscience, King's College London, London, United Kingdom, 2 London School of Hygiene & Tropical Medicine, London, United Kingdom, 3 Institute of Liver Studies, King's College Hospital NHS Foundation Trust, London, United Kingdom, 4 Faculty of Life Sciences and Medicine, School of Immunology and Microbial Sciences, King's College London, London, United Kingdom, 5 The Roger Williams Institute of Hepatology London, Foundation for Liver Research, London, United Kingdom, 6 NIHR Clinical Research Network South London, London, United Kingdom

* clair.le_boutillier@kcl.ac.uk

**Data Availability Statement:** This study uses data (containing potentially identifying and/or sensitive

## Abstract

### Objective

The success of pharmacological randomised controlled trials (RCTs) depends on the recruitment of the required number of participants. Recruitment to RCTs for patients with cirrhosis and small oesophageal varices raises specific additional challenges. The objectives of the study were 1) to explore patient perspectives on factors that influence RCT recruitment, 2) to understand factors that influence the success of recruitment from a staff perspective, and 3) to identify opportunities for tailored interventions to improve trial recruitment in this context.

### Methods

The qualitative study was embedded in a multi-centre blinded RCT (BOPPP trial) and was conducted alongside site opening. Semi-structured interviews were conducted with patients who enrolled to participate in the trial (n = 13), patients who declined to take part (n = 5), and staff who were responsible for recruiting participants to the trial (n = 18). An open approach to data collection and analysis was adopted and the Theoretical Domains Framework (TDF) was used to provide a theoretical lens through which to view influences on behaviour. Data was analysed using thematic analysis.

### Results

The findings consist of 5 overarching themes that outline trial recruitment influences at the patient, staff, team, organisational and trial levels: i) *patient risks and benefits* ii) *staff attitudes, knowledge and capacity*, iii) *team-based approach*, iv) *organisational context* and v) *Trial collective*. Patient-generated themes map onto thirteen of the fourteen TDF domains

information) collected from a small group of staff participants and a vulnerable patient population, and involves indirect identifiers (such as sex, ethnicity, location, etc.) that may risk the identification of study participants. Sharing data outside of the anonymised excerpts and quotations included in the paper will violate the agreement to which the participants consented. Data access queries may be sent to leedswest.rec@hra.nhs.uk.

**Funding:** This article presents independent research funded by NIHR Health Technology Assessment (17/32/04). The views expressed are those of the author(s) and not necessarily those of the NIHR or the Department of Health and Social Care.

**Competing interests:** The authors have declared that no competing interests exist.

and staff-generated themes map onto all TDF domains. The overarching themes are not mutually exclusive; with evidence of direct interactions between patient and staff-level themes that influence recruitment behaviours.

## Conclusions

This study uses a theory-informed approach to gain new insights into improving clinical trial recruitment for patients with cirrhosis and small oesophageal varices. Although people with cirrhosis often display decreased healthcare-seeking behaviours, we found that patients used research to empower themselves to improve their health. Pragmatic trials involving unpredictable populations require staff expertise in building trust, and a deep knowledge of the patient group and their vulnerabilities. RCT recruitment is also more successful when research visits align with what staff identified as the natural rhythm of care.

## Trial registration

ISRCTN10324656; https://clinicaltrials.gov/.

## Background

Liver disease is the fifth commonest cause of death in the developed world and is rising in incidence [1]. Liver cirrhosis causes portal hypertension and this in turn leads to the common complication of varices in the oesophagus, which are prone to haemorrhage [2], a very difficult experience for patients and clinicians. It is important that patients with oesophageal varices (OV) are monitored given their increased risk of variceal haemorrhage [2]. Monitoring is by endoscopic assessment of the size and appearance of the varices as both directly affect the risk of VH or death. Despite the advances of medical, endoscopic and radiological therapy the mortality rate from acute VH is 10%-20% [3]. Prevention of VH is therefore vital in those who have varices. The current evidence base and international recommendations identify a clear benefit in the reduction of VH with non-selective beta blockade (NSBB) in patients with moderate-large varices (>5mm in diameter), or those with advanced liver disease [4]. There is however currently no clear evidence to guide the use of NSBB in patients with small varices. The Beta-blockers Or Placebo for Primary Prophylaxis of oesophageal varices (BOPPP) trial is a phase IV, multi-centre Randomised Controlled Trial (RCT) that aims to determine the clinical and cost-effectiveness of using Carvedilol and placebo in the prevention of variceal bleeding for small OV in patients with cirrhosis. Patients are assessed for trial eligibility following detection of small OV at gastroscopy and those found to be eligible (all patients with small (grade 1) OVs due to cirrhosis of any cause) are invited to take part.

Recruiting patients with cirrhosis and small varices raises specific challenges. The patient group are clinically vulnerable because they are living with a chance of VH or death, and for those whose condition relates to alcohol use, there are additional vulnerabilities and risks. Clinical staff acknowledge the extent of the risks of living with cirrhosis and build relationships with patients and their families to support them. However, the influence of various factors such as stigma and lifestyle can, for some people with liver disease, contribute to difficulties engaging with health care, creating an additional complication of being hard-to-reach and limiting their involvement in research [5].

Given the condition of the target population, their possible history of traumatic oesophageal bleeding, and recruitment in the context of endoscopy and gastroscopy, the BOPPP trial faces a number of potential challenges to recruitment. Understanding the context of the trial and

the views of those with key perspectives on the experiences involved, will offer insights into potentially difficult and sensitive recruitment encounters. This is important as successful conduct of all RCTs depends on recruiting the required number of participants [6]. The challenge of trial recruitment is a long-established concern that is widely acknowledged, with factors that influence recruitment being well documented (i.e., additional demands of the trial, lack of staff and training, concerns about the impact on the clinical relationship, uncertainty about information and consent and lack of rewards and recognition) [7–10].

While studies on recruitment form a large part of qualitative research in trials, research that gathers the views of both staff and patients is important, because staff and patients offer different expertise and perspectives [11,12]. Conducting a qualitative study within the BOPPP trial (a trial that is aligned to standard of care visits and without obvious risks or demands on staff and patients) provides valuable learning and allows us to broaden the focus of recruitment barriers and facilitators across different system actors and system-levels. The broad aim of this qualitative study was to gain an understanding of the particular recruitment circumstances and conditions of the BOPPP trial, by gathering the views of staff and patients. The specific objectives of the qualitative study were to 1) to explore patient perspectives on factors that influence RCT recruitment, 2) to understand factors that influence the success of recruitment from the staff perspective, and 3) to identify opportunities for tailored interventions to improve trial recruitment.

## Methods

### Study design

BOPPP is a UK-wide RCT that aims to recruit 1200 participants to evaluate if NSBB reduces variceal haemorrhage in cirrhotic patients with small oesophageal varices (Trial registration: ISRCTN10324656). This study used qualitative methods to provide important insights into the complexities of trial recruitment, to identify barriers and facilitators to recruitment, and to inform the development of interventions to support recruitment [13,14]. To meet objective 1, semi-structured face-to-face or telephone individual interviews were conducted with patients who were identified as eligible to take part in the BOPPP trial, and who had been approached for trial participation. To meet objective 2, semi-structured telephone individual interviews were conducted with staff responsible for recruiting to the BOPPP trial to explore their recruitment experiences. To meet objective 3, interviews were used to generate detailed data which could be used to identify theory-informed interventions.

The Theoretical Domains Framework (TDF) was used to provide a theoretical lens through which to view influences on behaviour [15]. The TDF outlines 14 theoretical constructs, developed from a synthesis of psychological theories to understand the determinants of behaviour. Although the TDF was developed to understand the behaviour of health professionals, it has recently been extended to include other areas relevant to behaviour change such as patient behaviour [16]. Table 1 outlines the TDF domains and provides an illustration of how each construct might be used to explore factors that influence recruitment practices.

Staff and patients were recruited from 14 NHS Trusts across the United Kingdom, chosen to provide a mix of different regions and perceived levels of success in recruiting to the BOPPP trial. Participants were selected purposively (those who were eligible to take part in the BOPPP trial or those recruiting to the BOPPP trial) on the basis that they could offer a particular perspective on trial recruitment and are the target adopters of behaviours that influence the success of trial recruitment. However, heterogeneity was sought in the sample (for example, those who declined participation as well as those who enrolled on to the trial, and those who were successful and less successful in recruiting participants to the trial) so that convergence and

**Table 1. The theoretical domains framework and example application to recruitment.**

| DOMAIN | Application to recruitment practices |
|---|---|
| Knowledge | The level of knowledge and understanding of patients and staff on trial processes and procedures, intervention/treatment etc. that informs the decision whether to take part. |
| Skills | The skills of patients and staff to engage with/support and influence engagement with research. |
| Social/professional role and identity | The personal qualities of patients and staff that might influence recruitment practices. |
| Beliefs about capabilities | The beliefs of patients about their own ability to commit/participate in research. The beliefs of staff about their own ability to support others to commit/participate in research and the beliefs of staff about patients' ability to participate in research. |
| Optimism | The confidence of patients and staff that things will happen for the best or that desired goals will be attained by taking part in the research. |
| Beliefs about consequences | The beliefs of patients about the outcomes of taking part and the beliefs of staff about patients' taking part in research. |
| Reinforcement | Increasing the probability of a response by arranging a dependent relationship, or contingency, between the response and a given stimulus (i.e., participating in research provides additional monitoring and adds to care) |
| Intentions | A conscious decision to perform a behaviour or a resolve to act in a certain way |
| Goals | The goals of patients and staff that influence recruitment practices. |
| Memory, attention, and decision processes | The cognitive processes of patients and staff (memory, attention and decision making) that influence recruitment practices. For example, staff being able to remember and attend to prioritising research recruitment while working across a number of research studies. |
| Environmental context & resources | The environment of patients and staff (i.e., physical or resource factors) that influence recruitment practices. |
| Social influences | The social influences (i.e., interpersonal processes that can cause individuals to change their thoughts, feelings, or behaviours) of patients and staff that influence recruitment practices. |
| Emotion | The extent to which the emotion of patients and staff influences recruitment practices. |
| Behavioural regulation | Anything aimed at managing or changing objectively observed or measured actions (i.e., deliberate organisation of research tasks/time to reflect on the recruitment effort). |

divergence could be examined within the sample. Incentive payments, which were not considered unduly influential, were offered to those who declined to enter the trial to advance the goals of the study [17]. Participants were over 18, and proficient in English.

Eligible patients were first approached by the local research team (for example, local principal investigator, research nurse), and subsequently recruited to a telephone or face-to-face interview (depending on preference) by the lead author *via* the telephone or face-to-face. Potential staff participants were approached and recruited to a telephone interview by the lead author *via* the telephone or email. Written or verbal informed consent was obtained from each participant after a full explanation and information leaflet was given and time allowed for consideration. The right of each participant to refuse to participate without giving reasons was respected. All participants were free to withdraw from the study at any time without giving reasons and without prejudicing further treatment or employment. Written consent was not practical or appropriate for those participating in the research by telephone, so in these cases, verbal consent was taken by telephone with the conversation being audio-recorded. A witness [18] then verified consent after listening to the audio-recording. The consent procedures were granted ethical approval and recruitment continued until thematic saturation was reached [19]. The stages of recruitment are illustrated in S1 File.

## Data collection

Interviews used flexible open-ended questions for early data collection to gather a rich and detailed understanding of participants' perspectives. Patient interviews explored their views on trial recruitment including the acceptability of randomisation and other possible barriers and enablers associated with the proposed intervention and trial. Patients who declined to enter the trial were also asked for their reasons for declining participation. Staff were asked about their recruitment experiences including barriers and facilitators to recruiting patients with small oesophageal varices and cirrhosis. Prompts based on the TDF were used to consider factors related to the cognitive, affective, social and environmental influences on behaviour should they have relevance to the participants' narrative [16]. However, the TDF was used flexibly to allow a natural narrative flow to the interview and to enable participants to share information that was important to them [20]. A narrow approach to applying the TDF could have meant that contextual information around how wider determinants of behaviour interact would be lost. The patient and staff interview schedules were revised iteratively in response to the priorities and concerns of participants and are included in S2 File.

Interviews were conducted by telephone or face-to-face by the lead author (a post-doctoral qualitative research fellow in applied health and population research, with a clinical background in Occupational Therapy), lasted around 45 minutes, were audio-recorded and transcribed verbatim. Researcher reflexive notes were kept after each interview to consider the interaction with the participant, and to detail initial thoughts. Interviews were conducted by the lead author between September 2019 and September 2020.

## Data analysis

Initial inductive thematic analysis was used for data analysis, where analytical concepts and perspectives are derived from the data in a deliberate and systematic way [21]. This approach allows unexpected themes to be identified and does not restrict the investigation to predetermined concepts or prejudge the significance of concepts. In this way, the inductive approach ensured that non-TDF related factors were also identified [16].

Patient and staff data were initially analysed as separate groups. Data analysis began with individual-level case-by-case repeated re-reading of interview transcripts and re-listening of sound files for data immersion (CL). This was followed by line-by-line open coding, where individual extracts were coded under one or several themes to fully capture their meaning. Each theme was refined, and where data allowed, further sub-themes were developed. Thematic maps, visual representations of the themes, were used to organise the themes by clustering all codes according to connections in the data and by considering the patterns and relationships between themes. An initial coding frame for patients and for staff was developed and interpretive analysis was then undertaken to group the themes and sub-themes into overarching themes. Additional codes, refinements to the specifics of themes, and thematic patterns continued until theoretical saturation was achieved [21]. Thematic saturation occurred when the emergent themes had been fully explored and new data was easily accommodated within them. Comparisons between participant groups were explored and the two coding frames were integrated to provide the final coding framework. The themes were then mapped to the TDF domains to provide a theoretical understanding of factors that influence recruitment practices.

Data analysis occurred concurrently with data collection using NVivo QSR International qualitative analysis software (version 12). The lead author directed the analysis. Coding by a second rater [18] was undertaken to allow the opportunity to compare interpretations of data, to provide an opportunity to reflect on the coding approach, and to enhance rigour.

## Ethical approval

Ethical approval was obtained from the York and Humber (Leeds West) Research Ethics Committee and the Health Research Authority (REC reference 19/YH/0015).

## Results

### Participants

A total of 36 individual interviews were conducted with i) patients who entered the trial (n = 13), ii) patients who declined to enter the trial (n = 5), and iii) staff responsible for recruiting patients to the trial (n = 18). Patient characteristics are shown in Table 2 and staff characteristics are detailed Table 3.

**Table 2. Patient participant characteristics.**

| Trial Participation n (%) | Entered<br>n = 13 | Declined<br>n = 5 |
|---|---|---|
| **Type of Liver Disease** | | |
| Alcohol-related | 10 (76.9) | 2 (40.0) |
| Non-alcoholic fatty | 2 (15.4) | 1 (20.0) |
| Hepatitis C | 1 (7.7) | 0 (00.0) |
| Primary biliary cirrhosis | 0 (00.0) | 1 (20.0) |
| Autoimmune hepatitis | 0 (00.0) | 1 (20.0) |
| **Time since diagnosis** | | |
| >1 year | 4 (30.8) | 2 (40.0) |
| 1–4 years | 5 (38.4) | 2 (40.0) |
| 5–9 years | 2 (15.4) | 0 (00.0) |
| 10 years + | 2 (15.4) | 1 (20.0) |
| **Age** | | |
| 41–50 years | 3 (23.1) | 0 (00.0) |
| 51–60 years | 6 (46.1) | 3 (60.0) |
| 61–70 years | 2 (15.4) | 1 (20.0) |
| >70 years | 2 (15.4) | 1 (20.0) |
| **Gender** | | |
| Male | 7 (53.85) | 1 (20.0) |
| Female | 6 (46.15) | 4 (80.0) |
| **Ethnicity** | | |
| White British | 9 (69.2) | 5 (100.0) |
| White Irish | 1 (7.7) | 0 (00.0) |
| White Other | 2 (15.4) | 0 (00.0) |
| Black/Black British-Caribbean | 1 (7.7) | 0 (00.0) |
| **Marital status** | | |
| Single | 1 (7.7) | 1 (20.0) |
| Co-habiting | 2 (15.4) | 0 (00.0) |
| Married | 4 (30.7) | 3 (60.0) |
| Divorced | 5 (38.4) | 0 (00.0) |
| Widowed | 1 (7.7) | 1 (20.0) |
| **Highest Level of Education** | | |
| Full-time education (e.g., school) | 7 (53.8) | 3 (60.0) |
| Further education (e.g., sixth form college or equivalent) | 1 (7.7) | 2 (40.0) |
| Higher education (e.g., university) | 2 (15.4) | 0 (00.0) |
| Post-graduate qualification | 3 (23.1) | 0 (00.0) |
| **Employment Status** | | |
| Full-time paid work | 2 (15.4) | 0 (00.0) |
| Part-time paid work | 4 (30.7) | 0 (00.0) |
| Voluntary work | 1 (7.7) | 0 (00.0) |
| Unemployed | 3 (23.1) | 3 (60.0) |
| Retired | 3 (23.1) | 2 (40.0) |
| **Been involved in research before** | | |
| Yes | 3 (23.1) | 1 (20.0) |
| No | 10 (76.9) | 4 (80.0) |

**Table 3. Staff participant characteristics.**

| Recruiting staff n (%) | Staff n = 18 |
|---|---|
| **Core Profession** | |
| Nurse | 9 (50.0) |
| Hepatologist | 3 (16.6) |
| Gastroenterologist | 2 (11.1) |
| Medical trainee | 2 (11.1) |
| Research assistant | 1 (5.6) |
| Research delivery manger | 1 (5.6) |
| **Grade** | |
| Band 4 | 1 (5.6) |
| Band 5 | 2 (11.1) |
| Band 6 | 7 (38.95) |
| Band 7 | 1 (5.6) |
| Speciality registrar (doctor training grade) | 2 (11.1) |
| Consultant | 5 (27.65) |
| **Time since qualification** | |
| 0–2 years | 0 (00.0) |
| 2 years + -5 years | 5 (27.65) |
| 5 years + -10 years | 1 (5.6) |
| 10 years + -15 years | 5 (27.65) |
| 15 years + -20 years | 4 (22.3) |
| 20 years + -30 years | 1 (5.6) |
| 30 years + -40 years | 1 (5.6) |
| N/A | 1 (5.6) |
| **Time in current post** | |
| >6 months | 5 (27.65) |
| 7–12 months | 0 (00.0) |
| 13–24 months | 5 (27.65) |
| 25–36 months | 3 (16.4) |
| 37–48 months | 2 (11.1) |
| 49–60 months | 1 (5.6) |
| 8 years | 1 (5.6) |
| 12 years | 1 (5.6) |
| **Time in research** | |
| >6 months | 4 (22.3) |
| 7–12 months | 1 (5.6) |
| 13–24 months | 2 (11.1) |
| 25–36 months | 2 (11.1) |
| 37–48 months | 1 (5.6) |
| 49–60 months | 1 (5.6) |
| 6 years | 1 (5.6) |
| 7 years | 2 (11.1) |
| 8 years | 1 (5.6) |
| 10 years | 3 (16.4) |
| **Number of research studies currently working on** | |
| 1 | 1 (5.6) |
| 2 | 3 (16.4) |
| 3 | 2 (11.1) |
| 4 | 1 (5.6) |
| 5 | 1 (5.6) |
| 6 | 1 (5.6) |
| 7 | 2 (11.1) |
| 8 | 1 (5.6) |
| 9 | 0 (00.0) |
| 10 | 2 (11.1) |
| 12 | 1 (5.6) |
| 15 | 2 (11.1) |
| 20 | 1 (5.6) |

Patients were recruited from nine of the fourteen Trust sites involved in the qualitative study, and staff participants were recruited from 13 of the NHS Trusts. All staff participants identified research as integral to their professional role. Of the patients contacted by the researcher (CL), 72% agreed to take part in an interview (seven people declined) and 82% of staff participants agreed to take part (4 people declined participation). Reasons for non-participation included deteriorating health for patients, and lack of time.

## Overarching themes

The overarching themes to emerge from the data were: *patient risks and benefits*, *staff attitudes*, *knowledge and capacity*, *team-based approach*, *organisational context*, and *trial collective*. Participants' accounts described recruitment barriers and facilitators across system-levels so each of the themes can be mapped as i) patient, ii) staff, iii) team, iv) organisational or v) trial level (across organisations) influences. Because of space limitations, a summary of each overarching theme is provided, and the sub-themes are not elaborated in this article; the full coding framework is included in S3 File. Table 4 maps the themes to the TDF domains and offers an illustration of where staff and patient perspectives align and/or overlap, and where they are different. Patient and staff perspectives furthers our understanding because the staff-level themes overlap, are not mutually exclusive, and have a direct influence on the patient-identified themes. For example, staff *knowledge* and *skills* will influence what/how information about the trial is provided to patients before they can make the decision whether to take part. Alongside, *social influences* and *beliefs about patient capabilities* (i.e., patients who are invested in their health and actively involved in their care) are prioritised and staff judgement (*decision-making*) dictates who will be informed about the research.

## Theme 1: Patient risks and benefits

Patient and staff participants spoke about the trial processes and procedures as both a barrier and a facilitator. For some patients, a three-year commitment influenced their choice of whether to enrol. One patient who declined to enter the trial stated: *'It's just a lot to take on isn't it, agreeing to do it for three years'* (QR1001). However, another patient conceptualised the length of the trial as an enabler to research participation, and an opportunity to receive ongoing support: *It's three years, that three years you're being monitored'* (QR1102). Staff responsible for recruiting to the trial agreed that this was a positive consequence of participating in research and spoke about the reinforcement of additional monitoring, and that the opportunity to be involved in clinical trials adds to care.

Some patients spoke about the timing, and the influence of the point at which they found out about the trial. One patient commented that the research was *'sprung on me'* as it was mentioned at the same time as undergoing an endoscopy procedure:

> *'I didn't know anything about the trial until my second endoscopy, and at the time, there was a woman [research nurse] saying to me all about the trial, another guy [endoscopist] saying, 'stick your arm out here', another one going, 'lay back', And it all happened so quickly, I just came away with the information'*,
>
> (QR1002)

Another patient explained that the timing of the invitation to participate came too soon after their diagnosis, and they needed to give themselves priority to '*get everything back on track. . . I don't really want to rock the boat'* (QR0701). For some, all effort was on managing existing health needs, and the trial was another thing to worry about. However, other patients

**Table 4. Mapping themes and TDF domains.**

| Theme | Sub-theme | Patient-reported barriers | Staff-reported barriers | Patient-reported facilitators | Staff-reported facilitators | TDF Domains |
|---|---|---|---|---|---|---|
| Patient risks and benefits | Acceptability of the trial | Three-year commitment | | Three-years of ongoing support | Three-years of additional monitoring and research participation adds to care | Beliefs about consequences, emotion, reinforcement, optimism |
| | | Time for extra appointments | Time for extra appointments | | Aligning extra appointments with standard of care visits | Environmental context & resources |
| | | Misunderstanding the information | | Being supported to understand the information | Supporting patients to understand the information | Knowledge, emotion, beliefs about consequences, Memory, attention & decision processes Environmental context & resources, skills, knowledge |
| | | The timing of finding out about the trial—at what point in the system | | | | Environmental context & resources |
| | | | | Existing relationships with the team | Existing relationships with patients | Social influences |
| | An opportunity to help myself and others | | | Invested in my own health & a chance to be on treatment | A chance to be on treatment | Intentions, goals, beliefs about capabilities, behavioural regulation, beliefs about consequences, optimism, emotion |
| | | | | Regular clinic appointments and more timely treatment | Regular clinic appointments and more timely treatment | Reinforcement, beliefs about consequences |
| | | | | Reassurance to family and staff as 'seen to be helping self' | | Beliefs about consequences, behavioural regulation, social influences |
| | | | | Feeling valued and giving back to the clinical team | Feeling valued and giving back to the clinical team | Intention, social influence, reinforcement, emotion |
| | | | | Helping others and the greater good | | Beliefs about consequences, emotion |
| | Uncertainty and too much of a commitment | All effort is on managing existing health needs | All effort is on managing existing health needs | | | Emotion Beliefs about capability |
| | | Uncertainty about treatment, side effects, placebo | Uncertainty about treatment | | | Beliefs about consequences, emotion |
| Staff attitudes, knowledge & capacity | Staff attitudes | | BOPPP means asking more of patients | | Value of research & views on proposed intervention | Beliefs about consequences, Goals |
| | | | | | Views on trial set-up (i.e., opportunity to spend more time with patients) | Social/professional role & identity |
| | | | Views on eligibility criteria & predicting patient suitability | | | Social influences, memory, attention & decision processes |
| | Staff knowledge, experience & skills | | Knowledge of research processes, speciality, proposed intervention, eligibility criteria | | Knowledge of research processes, speciality, proposed intervention, eligibility criteria | Knowledge |
| | | | Confidence and ability to screen and identify patients | | Confidence and ability to screen and identify patients | Beliefs about capabilities |
| | | | Language barriers | | Communicate effectively and build a trusting relationship with patients | Skills, environmental context & resources |
| | Staff capacity | | BOPPP adds to workload | | | Environmental context & resources |

*(Continued)*

**Table 4.** (Continued)

| Theme | Sub-theme | Patient-reported barriers | Staff-reported barriers | Patient -reported facilitators | Staff-reported facilitators | TDF Domains |
|---|---|---|---|---|---|---|
| Team-based approach | Shared views and goals | | | | Team-level understanding and prioritisation of BOPPP | Knowledge, goals, social influences, behavioural regulation |
| | Team coordination of BOPPP tasks | | Patients may want to speak with their doctor before deciding whether to take part | | Deliberate organisation of BOPPP tasks | Environmental context & resources, behavioural regulation |
| | Effective team communication | | Research staff are not always embedded in clinical team | | Team time to reflect on the recruitment effort/ Team relationships & shared communication across teams | Environmental context & resources, behavioural regulation |
| | Team leadership | | | | PI involvement and support | Environmental context & resources, social/professional role & identity |
| Organisational context | Organisational culture and leadership | | | | Trial is well promoted and supported within organisation | Environmental context & resources, social/professional role & identity |
| | | | Amount of time available for endoscopies | | Degree to which other aspects of care delivery conflict or align with BOPPP | Environmental context & resources |
| | Organisational resources | | Limits on PI time | | | Environmental context & resources |
| | | | Travel & translator costs | | | Environmental context & resources |
| | | | Room availability | | | Environmental context & resources |
| | Organisational workflows | | Fewer than expected eligible patients | | Optimising local eligibility screening processes | Environmental context & resources |
| | | | Scheduling and timeframes—difficult to line up the ducks | | | Environmental context & resources |
| Trial collective | Feeling part of the BOPPP team across sites | | | | Links with chief investigator and communication with central team | Social influences, social/professional role & identity |

explained how involvement in the research would support their goal to be invested in their own health, for example, to give up alcohol, and offered them a chance to receive treatment:

> 'You don't know if you're on the placebo or the beta blocker... I understand perfectly well why you have you do that, You're not going to establish the efficacy of the treatment without doing it so that's fair enough, But if beta blockers help with the varices, then presumably, and I'm on the beta blockers, Then I'm going to get the benefit out of it,'

(QR1202)

Patients also described how the beliefs about consequences (i.e., a chance to be on treatment) and the behavioural regulation (i.e., giving up alcohol) of taking part was also a

reassurance for family. Others felt that by taking part in the research, and being seen to be helping themselves, they would receive better healthcare in the future if they ever needed it. Both staff and patients identified research participation as an opportunity for more regular appointments:

> *'Whereas there is obviously a slippage on normal appointments, simply because so many people are being diagnosed, and instead of sticking to a 12 month or a six-month appointment, those slip a month or two, Whereas if you're in the research study, they stick quite religiously to the time period simply because the research study needs to have its results when it needs to have its results, He said you'll get more timely treatment, and you won't have any slippage on your appointments,'*

> (QR1202)

However, both patients and staff identified additional appointments as a barrier to research participation. Staff explained how they attempt to limit the number of extra hospital visits by coinciding initial appointments with standard of care visits. One member of staff reported a difficulty engaging people '*if it's not within the natural rhythm of their treatment*' (QR0500). Other staff members explained that the ability to coordinate appointments also depends on environmental resources such as space and doctor availability. Staff also identified the length of time it requires to undertake the research appointment as a potential barrier to participation:

> *'I think baseline screening [is] quite intense, It's quite a lot of information to collect from the patient... we noticed that it did take quite a lot of time, especially when the patient has a lot of medications and perhaps other, like a history of certain illness or conditions and things to note down, It did take quite a while,'*

> (QR1200)

Some patients reported that they had understood the trial and what was expected of them and explained that the trial information had been presented in a clear and accessible way that had enabled them to make an informed decision about taking part. Other patients had a different understanding of the trial information or were worried about the possible consequences of side effects or the chance of placebo. One explained that it helped to have time to talk through the information with family, and that they agreed to take part after discussing the information with their son. Staff and patients felt that the information would be better understood if it could be communicated in simpler language and delivered in a face-to-face interaction with reassurance that ongoing support is available. This is particularly important given the vulnerabilities that could exist in the trial population of patients with cirrhosis, a condition associated for some of the participants with alcohol consumption. One patient made precisely this point:

> *'If it's explained in a better way, that it's not going to interfere with what's going on, and there are some benefits for people next down the line... I think people need to see a face to have that reassurance that we are going to mind you, we're looking after you, you'll still be under our wing... The key point is if you can get the information in a simpler language because, I'm not belittling anyone, but when you've drank as much as we drink, your brain cells aren't the best at retaining information, where a one-to-one conversation, it goes in, you remember... but somebody just giving you a pamphlet, it doesn't register,'*

> (QR1402)

Staff explained that, because of environmental context and resources, it is not always possible to meet patients face-to-face at the point of them being found to be eligible for the trial, and identified strategies to maximise success for research recruitment when 'cold calling':

*'It'll be beneficial if the consultant can mention the trial to the [patient] . . . so that I don't just call out of the blue, So, what I do now, I send them a letter, and then follow up with a call, so that it doesn't come as a shock to them,'*

(QR1000)

This also adds to the idea of creating rather than just adding on to *"a natural rhythm"* (QR0500). Some patients went on to explain that their view of the acceptability of the trial was influenced by the fact that they had an existing relationship with, and confidence in, the team providing the information about the trial:

*'I think by both knowing each other, they can give more to me and I can give them a lot more. . . Yes, [and] that [my consultant] mentioned about it, I happened to be there, so I said 'yes, just give us the info and we'll see', But only that I had that appointment, and if I had been with a different doctor, I may not have,'*

(QR1402)

This social influence was echoed by staff, who stated *'a lot of it is human interaction and familiarity, and these kinds of relationships' (QR0500)*. The experience of taking part in the research was described as a *'two-way street'* whereby it's about *'giving a little bit back'* as well as receiving personal gains (QR1402). Many of the patients who agreed to participate in the trial explained their decision was influenced by a desire to contribute, and to show appreciation for the clinical team by giving something back:

*'I just felt that when I was so poorly, and I was in hospital for ten weeks when I was first diagnosed, and they kept me alive, I just felt I owe them something, I need to put something back*

(QR0802).

*I just wanted to do anything that I could to help, payback time kind of thing because what he has done to save my life, if I could help him in any way with his work, then it was a way of saying thank you basically,'*

(QR1201)

This was echoed in staff recollections of patient's reasons for participation:

*'Let's face it, a lot of patients got liver disease because of alcohol intake, Those ones, actually we've got on board; They stop taking alcohol, They're turning their life around, and they're very committed, They want to do something, And the one gentleman, he was saying it's something to keep him going, to be part of trial,'*

(QR0800)

For others, participation provided an opportunity to help others and to be a part of future treatment: *'My thought pattern was that if I could do something that can help other people then*

*why wouldn't I do that*?' (QR0902). Patients stated that they felt valued being involved in the research. One reported: *'they certainly make a big fuss of you, and they're very thankful that you're doing it. . . and you feel as though you're doing something very worthwhile,'* (QR1202) Staff went on to explain that they too felt there were positive feelings for patients of being involved:

> *'They feel important, They feel the information they're giving us is making a difference, It makes them feel better. . . like the placebo effect of just being on a trial. . . At the end of the day if that's a positive. . . I think it's a great thing, Like if you make someone feel better, important, that you're interested in them,'*

> (QR0800)

### Theme 2: Staff attitudes, knowledge & capacity

The theme 'staff attitudes, knowledge and capacity' describes factors identified by staff that complement and sit alongside the patient data. Staff attitudes and their perceived value of research, as well as views on specific features of a clinical trial like the treatment being offered or trial design and conduct, influence the level of staff engagement with research. A consultant hepatologist explained the benefit of providing evidence for treatment and stated that the BOPPP trial has identified '*an important question for patients, and an important question for physicians*' (QR0500). Some staff argued that all patients should have access to research, while others reflected on the importance of the research, beliefs about consequences, and the opportunity that the trial provides for treatment:

> *'. . .the trial allows us to be able to approach these patients to say, we can offer you a treatment here, Obviously, you might get randomised the placebo, but we can offer you a treatment here that you otherwise wouldn't normally get,'*

> (QR0900)

Some staff went on to explain the value of the treatment; a research nurse with clinical experience in hepatology stated:

> *'I've looked after so many patients with oesophageal bleeds, To prevent them is everything, I've spent years looking after them, And when they come in when they're bleeding, it's pretty scary and you think, oh, why can't something be done about it, you know? They're usually vomiting blood or having really bad melena and I've had quite a lot of them die from that,'*

> (QR1200)

One research nurse explained that the phase of the clinical trial often influences the level of commitment from clinicians to support research because phase IV trials (like BOPPP) are *'not new, big and exciting'*:

> *'Yesterday I spoke to a consultant who will be working on a trial with us, and he said, 'well I just work on commercial trials', And I said, 'we've all done that, honey, But this, is aimed at changing practice, and it's looking at patients who normally fly under the radar and don't get looked at because they're not interesting',*

> (QR0800)

However, for others the intervention being offered in the trial was perceived as easy *'because it's actually already a tried and regular medication that you use for Grade 2 varices, So, it's not like a completely new drug that we don't know the side effects of or anything'* (QR0300). Other members of staff engaged positively because they were satisfied with the trial set-up:

*'I quite like BOPPP in the sense that you get a bit more time to spend with the patients... you get a bit more of a chance to screen patients properly, spend time with them, arrange appointments and pursue their needs,'*

(QR0900)

Staff views on patient eligibility and predicting patient suitability also influenced who they recruited: *'It's one thing, a patient being eligible, but you need to take into consideration whether they're suitable as well'* (QR0900). One staff member explained the priority to have 'good database' with fewer research participants who were likely to engage with the research for the duration of the trial:

*'I know we need the numbers for research, but we rather have good patients... but keep them for all the three years and have a good database,'*

(QR0801)

Another staff member explained:

*'Whenever you aim to recruit a fairly high number of patients, this is more like real life data... some of the patients do not turn up to their follow up appointment, do not turn up to their ultrasound scan appointment, and this is basically a real-life problem,'*

(QR0100)

Some staff explained that in their view it was important to recruit selectively, where patients who are active in their own care are prioritised for research and factors such as compliance with medication and appointment attendance are considered before approaching other eligible patients for participation:

*'Because, although someone might be eligible, it's whether they're suitable..., There's no point in recruiting somebody to a trial in which they need to take a medication for which, in the past or currently, that they don't take it, or they take it sporadically, when they want, When it's a trial, you have to know that they're compliant with it... [and] we don't want to recruit patients who will drop on us after one week, they don't come for one week appointment'*

(QR0801)

Staff across sites described research as *an extra hassle*, especially for patients who have other demands like caring responsibilities and work commitments (QR0600) and another stated, *'also people are quite reluctant, the chaotic lifestyle doesn't help* (QR1200). However, some staff spoke about the need to engage 'the other' demographic:

*'We've had other patients who we've spoken to about it with, and haven't responded to follow-up calls, or haven't made the one-week appointment from that demographic, I think we need to, as a site, think about strategies about how we engage a bit better with that group'*

(QR0500).

One staff member explained how a relationship of trust is necessary for patients to engage with research:

*'There's one patient I scoped last week who had grade one varices but was dependent on alcohol and was quite a sort of chaotic individual, was trying to cut down but was struggling, and struggled with clinic appointments, and was very anxious and I haven't ruled out inviting her completely, but even getting her on to the endoscopy was such a big deal. . . The trust is so fragile any way that you need to do one thing at a time,'*

(QR0500)

Some staff held the view that a commitment to trial participation meant asking more of patients. This was particularly challenging for those who felt there were risks to providing extra clinical input in stable disease, and where there is also chance of beta-blocker side effects on quality of life:

*'There are challenges in prescribing non-selective beta blockers because they can make people feel a bit rubbish, and so you need to know that you're doing it for good reason, You're often treating patients, particularly with this stage of disease, who are working, they're often young patients with family commitments, And you're prescribing a medication that can potentially affect energy levels, sleep, erectile function, things that matter to our patient cohort,'*

(QR0500)

Staff explained how knowledge, experience and skills, and the beliefs they have about their capabilities influence trial recruitment. In particular, staff knowledge of the eligibility criteria, the speciality, the proposed intervention, research processes influence the ability of staff to recruit to the trial. Alongside knowledge, confidence and ability to screen and identify patients and to communicate effectively and to be able to answer questions and provide accurate information was also highlighted as an influence. For example, screening was identified as 'relatively labour intensive', complicated when staff are not familiar with the speciality:

*'The screening has been quite tricky for our research coordinators because they're not specialist liver research coordinators, they're general research coordinators, They haven't done any liver work before, So, I'm doing a lot of the screening and going through the initial exclusion-inclusion criteria, just because they haven't learnt it. . . so it's mostly me because the research team don't currently feel confident enough to discuss the pros and cons of the medication, whereas obviously it's something I do at clinic all the time*

(QR0200)

Staff capacity, fitting BOPPP into existing workloads, and prioritising tasks when working on between one and fifteen studies were identified as recruitment influences. Some staff stated that they do not have protected time for different research studies they are working on:

*'We just balance it around our other work, Like, you know, screening and things like approaching patients, We don't have a specified time for it, no,'*

(QR0400)

Other staff work over multiple specialties and do not have the specific experience like dedicated liver research staff:

*'I'm very keen for us to gain experience and if we don't do these sorts of things, we'll never get the experience, but as a relatively inexperienced hepatology site, the research support staff find some of it quite challenging I would say... Part of the problem with our research staff is we're expected to work over multiple specialties, so it's not gonna be like [another site], who have got dedicated liver research staff... I got a list of people with small varices for them just to go through the basic exclusion criteria, and they did struggle with that a bit,'*

(QR0200)

### Theme 3: Team-based approach

A team-based approach influences the success of trial recruitment. The sub-themes to emerge from the data were: *shared views and goals, team coordination of research tasks, effective team communication*, and *team leadership,* One member of staff explained how shared goals influenced the ability to move recruitment efforts forward:

*'At the moment we are very lucky in that we have got a registrar who is on rotation, who is very keen to get patients for the trial and then spending a lot of time with me screening patients,'*

(QR0400)

A team-level understanding of the trial, and deliberate organisation and pre-planning research time also allows teams to identify which team members are responsible for which tasks:

*'I would say that BOPPP is one of the easier trials to be part of in terms of how much time it takes and how much you have to invest into it because, as I said, we just do that initial screening and baseline, I think one of the nurses usually follows them up in a week, titrate, and they get asked if they've got any questions, and then it's just follow up with consultants, so I think in a way that's a good thing about BOPPP... yeah, it's something that we can just bounce off each other, and if one of us can't do the one bit, then there's a couple of others around that can recruit, so I think it's working quite well probably so far,'*

(QR0300).

Staff also explained that staff familiarity, and the seniority of those recruiting influenced whether patients enrolled onto the trial:

*'When I speak to them about BOPPP, they say 'I want to speak to my doctor first'... it gives a bit more authority, and patients are more open to it...'*

(QR1000)

*'They're more likely to say yes to a doctor... He's in a better position to answer any questions that they have,'*

(QR1200)

Staff spoke about the benefit of shared communication across research and clinical teams and of research being integrated into clinical care:

*'So, it helps if, for example, [consultant] who would do [endoscopy], and he knows they're eligible, he would give them a patient information sheet, talk to them for a couple of minutes, and then we follow it up, So, in those cases I think it's a lot easier because they've already spoken to a consultant,'*

(QR0300)

Alongside, the value of teamwork and relationships with other departments like pharmacy and endoscopy influenced the success of recruitment:

*'Logistically getting everybody to work together has been the most beneficial part, And as I say because I'm not a nurse, we definitely need the PI to be available to discuss the clinical aspect, And having pharmacy prepared to dispense and such, it saves a lot of time for the patient as well,'*

(QR1200)

Team leadership and the support and availability of the local principal investigator (PI) also influenced staff ability to recruit. Staff called for support for local PIs and explained that research is often conducted in the PIs own time:

*'I think having a proactive investigator helps a lot, someone who's interested in actually recruiting the participants for us to be able to screen them and then potentially get them onto the trial, That's probably the biggest thing,'*

(QR0900)

*'. . .she's not getting paid for it and a lot of the other doctors do it on the side, and they're not getting paid. . . So, it's just finding time for them to do their own work, they've got busy clinics, [The PI's] been on the ward, doing the ward rounds as well, so she doesn't get much free time to then I blocked her email [laughs], I filled her inbox. . . It's just support for them, especially cos she's a new PI as well,'*

(QR1200)

Staff explained how PIs conduct research appointments in their own time. One PI stated:

*'All the patients I've seen, or I've recruited, they were additional time for me, in a way additional unpaid time for me, So, I do them either after the clinic time or in my spare time,'*

(QR1301)

## Theme 4: Organisational context

The environmental context and resources influence staff ability to recruit successfully. The sub-themes to emerge from the data were: *organisational culture and leadership*, *organisational resources*, and *organisational workflows*, Staff stated that it helps if the trial is well promoted and supported within the organisation. One staff member explained that the research had been promoted in local teaching sessions:

*'[The PI] runs the teaching session on Friday with the registrar, so she's mentioned about the trial to the doctors, So that helps,'*

(QR0201)

For some staff, having available clinic space for research appointments is a resource requirement:

*'We have to book a clinical room to see the patient in, which is sometimes a bit of an obstacle here, They're a bit short on space,'*

(QR1201)

Other sites have access to a clinical research facility (CRF):

*'We've got a separate CRF unit, so we try and get [consultants] to come here to see the patients, rather than where at a clinic it can be rushed... I think, at clinic, it can be difficult for the investigators to spend as much time as they can with the patient... they'll only get a certain amount of time per clinic appointment, so the investigators have to rush through consent etc, and then, also, we see [the patients] after that,'*

(QR0504)

Organisational workflows and variability in local eligibility screening processes influences the success of recruitment. For some, the limited number of endoscopy time reduced the chance of identifying eligible patients. For others, the challenge of working with large numbers of endoscopists was informing everyone about the study and being available: *'there's so many lists that we can't be there'* (QR0400).

In terms of eligibility, staff explained how 'there's been probably over half that I found that have just been outside the window. (QR0901)

*'When we went through our data, it showed a lot with varices, but because they tend to be patients with stable disease, they have their one-year scope booked, or the six-month clinic ultrasound booked, it was quite difficult to line up the ducks,'*

(QR0500)

## Theme 5: Trial collective

Staff requested communication and involvement with other trial sites and reported an interest in becoming part of a wider trial team across sites, indicating a desire for a collegiate approach. It was suggested that the central trial team might develop opportunities to link across sites. Staff also spoke about existing relationships with the central team and the value of having direct interactions with the chief investigator:

*'[The chief investigator] came here a couple of weeks ago, So, he actually came to site, he was giving a lecture, so he popped in, I was able to go through lists of names and we talked about the beta blocker issue,'*

(QR1200)

## Tailored interventions to improve BOPPP recruitment

We mapped the themes to the TDF domains and drew on existing literature to inform our response, and identified 16 strategies that could be used to support recruitment by addressing

the barriers and enhancing the enablers to recruitment. The research team met to discuss the feasibility and practicality of offering each recruitment strategy and generated a priority list for action. Table 5 outlines the themes and corresponding recruitment strategies.

## Discussion

The findings consist of five overarching themes: i) *patient risks and benefits*, ii) *staff attitudes, knowledge, and capacity*, iii) *team-based approach*, iv) *organisational context* and v) *trial*

**Table 5. Themes and corresponding recruitment strategies.**

| Identified Individual-level Characteristics | | |
|---|---|---|
| **Characteristics & Description** | **Recruitment strategy** | **Aim** |
| **Staff attitudes** | | |
| *The value of BOPPP* Degree to which the value of BOPPP (and a phase 4 trial) is perceived by staff | • Central Team call-in<br>• BOPPP newsletter<br>• Video updates<br>• Training resources on website/integrate into existing local teaching sessions | To proactively support individuals to identify the value of, and prioritise BOPPP.<br>To learn from others i.e., others have said, BOPPP adds to care, will provide evidence for treatment and is an 'easy' intervention. |
| *Providing extra clinical input in stable disease* Degree to which staff are concerned about providing extra clinical input in stable disease and the impact of possible side effects on QoL | • Video updates e.g., of chief investigator and chief scientific investigator addressing findings from the BOPPP qualitative interviews/practice experience | To address concerns of staff re: challenges of prescribing NSBBs in this patient group.<br>[NB e.g., to also address recent changes in NSSB use] |
| *Asking more of patients* Degree to which staff members perceive BOPPP visits to impact patients | • Video updates | To address concerns of staff re: challenges of asking more of patients who are living asymptomatically (i.e., time off work, travel for BOPPP visits). |
| *Predicting patient Suitability* Degree to which staff perceive or judge patients to be suitable for BOPPP (i.e., patients invested in own health Vs research is 'a hassle') | • Central Team call-in<br>• Training resources on website/integrate into existing local teaching sessions | To identify all eligible patients.<br>To reflect on how to engage those considered the 'other' demographic e.g., those of NFA, too old, with caring responsibilities—as well as those with more 'chaotic' lifestyles.<br>To consider how we make research less of 'a hassle' for patients. |
| **Staff knowledge** | | |
| *(Perceptions of) knowledge of eligibility criteria and ability to screen and identify patients* Degree to which staff are confident and skilled to screen and identify eligible patients | • Training e.g., optimal screening process outlined on SIV slides/BOPPP website<br>• CPD accreditation points available for all training<br>• Central Team call-in for trouble shooting and to provide ongoing consultation | To enable staff to be confident and skilled to screen and identify eligible patients.<br>To support staff to reflect on current screening processes and to consider optimal practices.<br>To provide an opportunity for staff to share challenges and to learn from each other's successes.<br>To provide learning around specific staff concerns re: ability to screen eligible patients when liver disease is not their speciality, new to research.<br>To offer a centrally-led strategy/opportunity for teams to liaise/share queries with the central team—and so sites feel a part of the wider team. |
| *Communication with eligible patients* Degree to which staff can provide information that is accessible to patients, and to answer questions about BOPPP | • BOPPP newsletter e.g., success stories that demonstrate effective communication with patients<br>• Video of patient/recruiter interaction | To learn from success i.e., some sites have shared success stories that have involved communicating with family members and patients.<br>[NB patients have declined to enter the trial because they have mis-understood the information].<br>To provide an opportunity to observe/'shadow' and learn from others. |
| **Staff capacity** | | |
| *BOPPP is more work* Degree to which BOPPP recruitment and research visits add to individual workload | • Central Team call-in<br>• Local BOPPP meetings and team reflection | To provide a space to consider the amount of work required for each recruited patient and how this fits into existing workloads (especially for clinical staff). |

*collective*. Each of the themes can be largely mapped onto dealing with i) patient, ii) staff, iii) team, iv) organisational or v) trial level influences pointing to the need to develop a whole-systems approach to recruitment. Individual staff, local teams (i.e., shared views and goals, team communication and co-ordination of tasks), and the support of the organisation (i.e., work-flows, resources, culture, and leadership) influence the success of RCT recruitment. Identified patient-level barriers and facilitators (i.e., finding out about the trial, timing) outline the attention and support required for patients, and *trial collective* (i.e., feeling a part of the team across sites) acknowledges the attention and support required for staff to enable successful recruitment. Although support that optimises patient participation in trials is well documented [22], this identification of higher-level staff attention and support (across organisations) is new.

While the findings can be mapped to the TDF domains, and compliment and align with existing literature on factors that influence RCT recruitment, knowledge is also extended by identifying the importance of the overall trial context, that is the specific population, condition and/or treatment being offered [23]. To optimise recruitment, trial teams must consider their context, and understand and respond to their patient population and condition, accounting for specific vulnerabilities that the patient group may have. In the case of the BOPPP trial, the patient group is dominated by alcohol-related cirrhosis with a risk of variceal bleeding, some-times leading to fatal consequences for patients. Our findings identify trust as a non-TDF domain and demonstrate that the context of a caring, existing clinical relationship supports successful recruitment; staff are invested, responsive and aware of the specific risks and want to find solutions for patients. This finding challenges previous concerns about the impact of research participation on the clinical relationship. Pragmatic trials, that is trials designed to evaluate the effectiveness of interventions in real-life practice, especially those involving unpre-dictable populations require staff expertise in building trust, and a deep knowledge of the patient group and their vulnerabilities. A collegiate approach to trial recruitment is essential to support staff in their endeavour [24].

RCT recruitment is also more successful when research visits align with what a staff member identified as the natural rhythm of care. If a trial is introduced when the patient doesn't have capacity to think about it (i.e., too soon after diagnosis) or asks the patient for extra action when it is burdensome (i.e., managing health needs), there is disruption to the natural rhythm. Therefore, the rightness of timing in terms of environmental context and resources as well as when people are receptive to information, to engagement and to action (beliefs about conse-quences/capabilities/emotion) impacts the success of recruitment [25]. For the BOPPP trial, some patients were eager for information and action, and it fitted with their need to do some-thing for themselves and/or for others. Stigma is common among patients with cirrhosis, often resulting in decreased healthcare-seeking behaviours [26]. However, our findings challenge this notion, and suggest that in some situations, where people might perceive that blame and shame are attached to their condition, a feeling of helping oneself, as well as being seen to help oneself provides an impetus for being involved in research. Where this is possible, patients who take part in research feel valued and staff value having more time to spend with patients. Alongside this, ongoing feedback and communication enables patients to remain involved (also supporting trial retention). Previous research also identifies a willingness to help others and to contribute towards furthering medical knowledge as reasons to be involved in partici-pating in trials [7].

Despite this, not all patients are informed about research opportunities. Staff *goals* and views on the value of research, that is, a patient's right to be informed of research opportuni-ties, influence their priority of informing patients about research opportunities. Staff *knowl-edge* and understanding (to provide a correct explanation of trial processes and rationale), the *environmental context and resources* (such as having time to be available at clinic appointments

and being mindful of at what point in the system the information is given), and *social influences* (such as having existing relationships with patients) influence the ability and *skill* of staff to support patients to understand the trial information (thereby supporting patient *memory*, *attention*, *decision processes*) before patients can make a decision whether or not to take part. Staff *beliefs of consequences* for patients (for example, impact on quality of life (positive or negative) and *of patient capabilities* (for example, their views of the ability of patients to engage with research when managing existing health needs) will determine whether they inform patients about the research and give them the opportunity to participate. Alongside eligibility, patient suitability is determined by staff. This is an important insight, because on the one hand, being involved in research is a right of care, and benefits all patients by involving them more directly in their own care, yet on the other hand those patients who are already well-engaged and active in their own care are prioritised and approached for recruitment. The opportunity to benefit from research should be open to all which will mean challenging implicit biases, taking time with recruitment, and possibly accepting a lower retention rate. Pragmatic results are needed to capture the impact for all patients, and not just those who are considered suitable, to reflect the real-world unbiased patient population we are trying to evaluate and serve. The focus on suitability in terms of selection strategy to improve study adherence e.g., *a good database* also places emphasis on patients in terms of the failure of potential retention and not on trial design or the imbalance of power in making the decision of whether or not to take part.

While the use of the TDF is established in healthcare implementation projects, its use in research environments is novel, the implications for using the TDF to inform behaviour change interventions in trial recruitment is new, and could have wider implications in other RCTs [27]. The findings informed the development of a recruitment strategy to support improvements in existing recruitment practices and addressed the need for additional behaviour change interventions to optimise recruitment to the BOPPP trial [28]. Interventions were proposed at individual staff (training, newsletter, video updates e.g., to address concerns of staff re: challenges of asking more of patients who are living asymptomatically or of prescribing NSBBs in this patient group), team (central team call-in (fortnightly contact between the lead trial team and each local site to forge links and support the development of individual-site BOPPP team meetings)), organisational (NIHR PI Associate Scheme to support leadership and develop and up-skill sub-investigators) and across sites (regional BOPPP meetings (linking with CRN network regions)) and BOPPP-wide study days (to create a learning collaborative, share best practices and tips, to network, and provide an opportunity for reflection across sites).

## Strengths and limitations

While the paper extends previous research by exploring patient and recruiting staff perspectives and experiences, it is important to note that the findings are specific to the BOPPP trial and are not empirically generalisable. It is possible, however, to enhance transferability by describing the research context and assumptions, and by making connections between the analysis of participants accounts and claims in the extant literature. It is also important to note, that despite focused efforts to purposively recruit patient participants from minority ethnic backgrounds, experiences of those from these groups are underrepresented. This recruitment difficulty is echoed in literature where it is reported that those from ethnic minorities are less likely to participate in research [29]. While we were unable to specifically explore ethnic variations and barriers to trial recruitment, we found that researcher bias and judgement of patient suitability dictated those who are prioritised for recruitment. Another possible reason for non-participation is the risk that asking patients to take part in an additional qualitative

study may inadvertently alienate them from their enrolment/retention in the clinical trial [30]. Staff also reported that the quality of relationships with patients is sometimes impacted by language barriers. Future strategies such as availability of interpreters, linguistic and ethnic matching, and an awareness of cultural practices and norms provide possible options for addressing these barriers [31].

Telephone interviews have received criticism for compromising interviewer/participant rapport and interaction, and for limiting contextual data due to the absence of face-to-face contact and visual cues [32]. However, this method of data collection is convenient, in that it is flexible (in terms of time, location), and allows for a wide reach (e.g., accessing BOPPP sites across the UK). This method was also invaluable in the context of the COVID-19 pandemic where telephone interviews were conducted during lockdown with patients classed as clinically extremely vulnerable. The anonymity of telephone use can also allow participants to disclose sensitive information and there is no evidence to suggest that they produce lower quality data [33]. A strength of the study is the thorough and systematic application of qualitative methods, including triangulation with staff and patient perspectives, and the use of the TDF to inform data collection, analysis, and theoretically driven recruitment strategies. This qualitative study provides support for the comprehensiveness and inclusivity of the TDF, demonstrating its value in use with patient/public populations in research environments exploring influences and determinants of behaviour.

## Conclusion

The study used a theory-informed approach to gain new insights into improving clinical trial recruitment for patients with cirrhosis and small oesophageal varices by mapping the findings to the TDF domains. The TDF provides a useful, flexible framework for identifying influences on patient and staff recruitment behaviours. A whole-systems approach to recruitment is required with a focus on the overarching trial context to ensure patient and staff support needs are met and recruitment is optimised.

## Supporting information

**S1 File. BOPPP stages of qualitative research recruitment.**
(DOCX)

**S2 File. Interview schedules.**
(DOCX)

**S3 File. Coding framework.**
(DOCX)

## Acknowledgments

We would like to thank all patients and staff who participated in this study and who generously gave their time and honest thoughts. We are very grateful to the local BOPPP research teams that helped to recruit participants and supported the qualitative study.

## Author Contributions

**Conceptualization:** Claire Snowdon, Vishal Patel, Mark McPhail, Christopher Ward, Vanessa Lawrence.

**Data curation:** Clair Le Boutillier, Mark McPhail, Ane Zamalloa.

**Formal analysis:** Clair Le Boutillier, Vanessa Lawrence.

**Funding acquisition:** Vishal Patel, Mark McPhail, Ben Carter, Vanessa Lawrence.

**Investigation:** Clair Le Boutillier, Christopher Ward.

**Project administration:** Ruhama Uddin, Ane Zamalloa.

**Writing – original draft:** Clair Le Boutillier, Claire Snowdon.

**Writing – review & editing:** Clair Le Boutillier, Claire Snowdon, Vishal Patel, Mark McPhail, Christopher Ward, Ben Carter, Ruhama Uddin, Vanessa Lawrence.

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
