## [Editor Report · Decision Letter 0]

18 Oct 2021

PONE-D-21-22188Using a theory-informed approach to explore patient and staff perspectives on factors that influence clinical trial recruitment for patients with cirrhosis and small oesophageal varicesPLOS ONE

Dear Dr. Le Boutillier,

Thank you for submitting your manuscript to PLOS ONE. After careful consideration, we feel that it has merit but does not fully meet PLOS ONE’s publication criteria as it currently stands. Therefore, we invite you to submit a revised version of the manuscript that addresses the points raised during the review process.

We look forward to receiving your revised manuscript.

Kind regards,

Tariq Jamal Siddiqi

Academic Editor

PLOS ONE

2. When reporting the results of qualitative research, we suggest consulting the COREQ guidelines: http://intqhc.oxfordjournals.org/content/19/6/349. In this case, please consider including more information on the number of interviewers, their training and characteristics; and please provide the interview guide used.

3. Please provide additional details regarding participant consent. In the ethics statement in the Methods and online submission information, please ensure that you have specified whether consent was written or verbal/oral. If consent was verbal/oral, please specify: 1) whether the ethics committee approved the verbal/oral consent procedure, 2) why written consent could not be obtained, and 3) how verbal/oral consent was recorded. If your study included minors, please state whether you obtained consent from parents or guardians in these cases. If the need for consent was waived by the ethics committee, please include this information.

“We would like to thank all patients and staff who participated in this study and who generously gave their time and honest thoughts. We are very grateful to the local BOPPP research teams that helped to recruit participants and supported the qualitative study. This article presents independent research funded by NIHR Health Technology Assessment (17/32/04). The views expressed are those of the author(s) and not necessarily those of the NIHR or the Department of Health and Social Care.”

“This article presents independent research funded by NIHR Health Technology Assessment (17/32/04). The views expressed are those of the author(s) and not necessarily those of the NIHR or the Department of Health and Social Care.”

7. Please upload a copy of Figure 1, to which you refer in your text on page 7. If the figure is no longer to be included as part of the submission please remove all reference to it within the text.

Additional Editor Comments (if provided):

Boutillier et al. performed a study, “Using a theory-informed approach to explore patient and staff perspectives on factors that influence clinical trial recruitment for patients with cirrhosis and small oesophageal varices” in which they assess the patients and staffs perspectives on factors that influence the recruitment of patients in the clinical trial, and they show that theory informed approach provides a useful, flexible framework for identifying influences on patient and staff recruitment behaviors. In my opinion, this study can be further improved by incorporating the following point:

1. More points regarding the inclusion and exclusion criteria of the trial will enhance the study.

2. It can be beneficial in mentioning what kind of questions were asked in interview along with mentioning the results of the participants.

3. Mentioning more personal comments from the patients or staff or team members regarding individual subthemes can enhance the study further.
---

## [Author Response · Author response to Decision Letter 0]

16 Nov 2021

We have removed the reference to Figure 1 from the manuscript for clarity. The stages of recruitment are illustrated in Supplementary information 1 (S1) (attached as a separate file).

We have checked the permissions regarding data sharing, and have amended our data availability statement: 

This study uses data (containing potentially identifying and/or sensitive information) collected from a small group of staff participants and a vulnerable patient population, and involves indirect identifiers (such as sex, ethnicity, location, etc.) that may risk the identification of study participants. Sharing data outside of the anonymised excerpts and quotations included in the paper will violate the agreement to which the participants consented. 

Thank you very much for reviewing this manuscript. Please find responses to each point raised by the academic editor and reviewer(s) below. All changes to the manuscript are highlighted with track changes. 

We have checked PLOS ONE’s style requirements and have changed the supplementary information file names to meet the requirements. 

2. When reporting the results of qualitative research, we suggest consulting the COREQ guidelines. In this case, please consider including more information on the number of interviewers, their training and characteristics; and please provide the interview guide used.

We have used the COREQ guidelines to report the study – and re-attach the checklist. We have included information on the number of interviewers and the two interview schedules are included in supplementary information 2. We have highlighted where this information is available in the text using bold. We have extended the information to include the training and characteristics of the interviewer (page 7).

3. Please provide additional details regarding participant consent. In the ethics statement in the Methods and online submission information, please ensure that you have specified whether consent was written or verbal/oral. If consent was verbal/oral, please specify: 1) whether the ethics committee approved the verbal/oral consent procedure, 2) why written consent could not be obtained, and 3) how verbal/oral consent was recorded. 

We have included additional information regarding participant consent – and clarified that patients were offered face-to-face or telephone interviews. We have confirmed that ethical approval was obtained to allow for written and verbal consent and have outlined the procedures (page 7).

“We would like to thank all patients and staff who participated in this study and who generously gave their time and honest thoughts. We are very grateful to the local BOPPP research teams that helped to recruit participants and supported the qualitative study. This article presents independent research funded by NIHR Health Technology Assessment (17/32/04). The views expressed are those of the author(s) and not necessarily those of the NIHR or the Department of Health and Social Care.”

We have removed all funding information from the manuscript and ensured the funding information in the Funding Statement section of the online submission form is correct.

“This article presents independent research funded by NIHR Health Technology Assessment (17/32/04). The views expressed are those of the author(s) and not necessarily those of the NIHR or the Department of Health and Social Care.”

Thank you for bringing this to our attention. We would like the data availability statement to read: 

The data that supports the findings of this study are available from the corresponding author [CL] upon reasonable request.

We have checked the manuscript and the ethics information is only included in the methods section. 

7. Please upload a copy of Figure 1, to which you refer in your text on page 7. If the figure is no longer to be included as part of the submission please remove all reference to it within the text.

Figure 1 has now been uploaded as supplementary information file 1.

8. Please include captions for your Supporting Information files at the end of your manuscript, and update any in-text citations to match accordingly. Please review your reference list to ensure that it is complete and correct. Any changes to the reference list should be mentioned in the rebuttal letter that accompanies your revised manuscript. If you need to cite a retracted article, indicate the article’s retracted status in the References list and also include a citation and full reference for the retraction notice.

We have included captions for the supplementary files (S1-S3) at the end of the manuscript. We have reviewed the reference list to ensure it is complete and correct. We have not made any changes to the reference list. 

9. More points regarding the inclusion and exclusion criteria of the trial will enhance the study.

This information has now been included (page 3).

10. It can be beneficial in mentioning what kind of questions were asked in interview along with mentioning the results of the participants.

We have outlined patient questions in the text and have highlighted where this information is available in the text using bold. We have added information about the kind of questions asked in staff interviews (p.7). The interview schedules are included in Supplementary information file 2. 

11. Mentioning more personal comments from the patients or staff or team members regarding individual subthemes can enhance the study further.

Because of space limitations, the sub-theme findings have been reported within the overarching theme. For example, one sub-theme of patient risks and benefits is acceptability of the trial. This is illustrated by data that explains how a three-year commitment influenced patients’ choice of whether to enrol in the trial (p.17). Another example is the perceived value of BOPPP and views to research as a sub-theme of staff attitude (page 20-21).

We hope that these responses and changes make the paper suitable for publication in PLOS ONE.

---

## [Decision Letter · Decision Letter 1]

23 Dec 2021

PONE-D-21-22188R1Using a theory-informed approach to explore patient and staff perspectives on factors that influence clinical trial recruitment for patients with cirrhosis and small oesophageal varicesPLOS ONE

Dear Dr. Le Boutillier,

Thank you for submitting your manuscript to PLOS ONE. After careful consideration, we feel that it has merit but does not fully meet PLOS ONE’s publication criteria as it currently stands. Therefore, we invite you to submit a revised version of the manuscript that addresses the points raised during the review process.

We look forward to receiving your revised manuscript.

Kind regards,

Tariq Jamal Siddiqi

Academic Editor

PLOS ONE

Journal Requirements:

Reviewers' comments:

Reviewer's Responses to Questions

**Comments to the Author**

1. If the authors have adequately addressed your comments raised in a previous round of review and you feel that this manuscript is now acceptable for publication, you may indicate that here to bypass the “Comments to the Author” section, enter your conflict of interest statement in the “Confidential to Editor” section, and submit your "Accept" recommendation.

Reviewer #1: (No Response)

2. Is the manuscript technically sound, and do the data support the conclusions?

Reviewer #1: Yes

3. Has the statistical analysis been performed appropriately and rigorously? 

Reviewer #1: I Don't Know

4. Have the authors made all data underlying the findings in their manuscript fully available?

Reviewer #1: Yes

5. Is the manuscript presented in an intelligible fashion and written in standard English?

Reviewer #1: Yes

6. Review Comments to the Author

Reviewer #1: Providing the best possible approach for improving RCT recruitment for patients with cirrhosis and small esophageal varices is of great importance. This is an excellent question, where theory-informed approach will provide new insights into improving trial recruitment for patients with cirrhosis and small esophageal varices. Analysis is well done and manuscript reads well.

I have several suggestions to the current version of manuscript

1. The authors should address issue of cultural factors and recruitment of ethnically diverse participants. This is critical issue; cultural and ethnic barriers might need more considerations. If data is not available on diverse participants, authors can use articles from literature to discuss more about this issue in the discussion section.

2. Similarly, authors should also discuss the issue of relationship between recruiter and study participants, which could be impacted by lingual, cultural and ethnic barriers in more culturally and ethnically diversified trials

3. Word “Registrar” in Table 3 may need more clarity for international readers.

4. Education levels in Table 2 are categorized as “Full-time education”, “Further education”, “Higher education”. “Post-graduate education”. These names of classification read vague, examples within bracket in fact provide more helpful information. Renaming each of levels of literacy may help bringing more clarity.

7. PLOS authors have the option to publish the peer review history of their article (what does this mean?). If published, this will include your full peer review and any attached files.

Reviewer #1: No

---

## [Author Response · Author response to Decision Letter 1]

13 Jan 2022

Thank you very much for reviewing this manuscript. Please find responses to each point raised by the reviewer(s) below. All changes to the manuscript are highlighted with track changes. 

1. Has the statistical analysis been performed appropriately and rigorously? 

We have conducted a qualitative study and have therefore not performed any statistical analysis. 

2. The authors should address issue of cultural factors and recruitment of ethnically diverse participants. This is critical issue; cultural and ethnic barriers might need more considerations. If data is not available on diverse participants, authors can use articles from literature to discuss more about this issue in the discussion section.

We have included this information in the Limitations section of the Discussion – and below for your reference (page 33). 

It is also important to note, that despite focused efforts to purposively recruit patient participants from minority ethnic backgrounds, experiences of those from these groups are underrepresented. This recruitment difficulty is echoed in literature where it is reported that those from ethnic minorities are less likely to participate in research (Smart & Harrison, 2016). While we were unable to specifically explore ethnic variations and barriers to trial recruitment, we found that researcher bias and judgement of patient suitability dictated those who are prioritised for recruitment. Another possible reason for non-participation is the risk that asking patients to take part in an additional qualitative study may inadvertently alienate them from their enrolment/retention in the clinical trial (Liu et al, 2011). Staff also reported that the quality of relationships with patients is sometimes impacted by language barriers. Future strategies such as availability of interpreters, linguistic and ethnic matching, and an awareness of cultural practices and norms provide possible options for addressing these barriers (Masood et al, 2019).

3. Similarly, authors should also discuss the issue of relationship between recruiter and study participants, which could be impacted by lingual, cultural and ethnic barriers in more culturally and ethnically diversified trials. 

Thank you for this important point. We have added the need to the impact of the recruiter-study participant relationship (text above) in the discussion (page 33).

4. The word “Registrar” in Table 3 may need more clarity for international readers.

Thank you for bringing this to our attention. A speciality registrar is a doctor who is working as part of a training programme in the UK. We have added this definition in brackets for clarification (page 11).

5. Education levels in Table 2 are categorized as “Full-time education”, “Further education”, “Higher education”. “Post-graduate education”. These names of classification read vague, examples within bracket in fact provide more helpful information. Renaming each of levels of literacy may help bringing more clarity. 

We have added examples within brackets next to each category label (Page 10). 

6. When reporting the results of qualitative research, we suggest consulting the COREQ guidelines. In this case, please consider including more information on the number of interviewers, their training and characteristics; and please provide the interview guide used.

We have used the COREQ guidelines to report the study – and re-attach the checklist. We have included information on the number of interviewers and the two interview schedules are included in supplementary information 2. We have highlighted where this information is available in the text using bold. We have extended the information to include the training and characteristics of the interviewer (page 8).

---

## [Editor Report · Decision Letter 2]

18 Jan 2022

Using a theory-informed approach to explore patient and staff perspectives on factors that influence clinical trial recruitment for patients with cirrhosis and small oesophageal varices

PONE-D-21-22188R2

Dear Dr. Le Boutillier,

We’re pleased to inform you that your manuscript has been judged scientifically suitable for publication and will be formally accepted for publication once it meets all outstanding technical requirements.

Kind regards,

Tariq Jamal Siddiqi

Academic Editor

PLOS ONE
---

## [Editor Report · Acceptance letter]

26 Jan 2022

PONE-D-21-22188R2 

Using a theory-informed approach to explore patient and staff perspectives on factors that influence clinical trial recruitment for patients with cirrhosis and small oesophageal varices 

Dear Dr. Le Boutillier:

I'm pleased to inform you that your manuscript has been deemed suitable for publication in PLOS ONE. Congratulations! Your manuscript is now with our production department. 

Kind regards, 

on behalf of

Dr. Tariq Jamal Siddiqi 

Academic Editor

PLOS ONE